# Quantification of All-Trans Retinoic Acid by Liquid Chromatography–Tandem Mass Spectrometry and Association with Lipid Profile in Patients with Type 2 Diabetes

**DOI:** 10.3390/metabo11010060

**Published:** 2021-01-19

**Authors:** Jakob Morgenstern, Thomas Fleming, Elisabeth Kliemank, Maik Brune, Peter Nawroth, Andreas Fischer

**Affiliations:** 1Department of Internal Medicine I and Clinical Chemistry, University Hospital Heidelberg, Im Neuenheimer Feld 410, 69120 Heidelberg, Germany; thomas.fleming@med.uni-heidelberg.de (T.F.); elisabeth.kliemank@med.uni-heidelberg.de (E.K.); maik.brune@med.uni-heidelberg.de (M.B.); Peter.nawroth@med.uni-heidelberg.de (P.N.); a.fischer@dkfz.de (A.F.); 2German Center for Diabetes Research (DZD), 85764 Neuherberg, Germany; 3Division Vascular Signaling and Cancer (A270), German Cancer Research Center (DKFZ), 69120 Heidelberg, Germany; 4European Center for Angioscience, Medical Faculty Mannheim, Heidelberg University, 68167 Mannheim, Germany

**Keywords:** retinoic acid, liquid chromatography–tandem mass spectrometry, hyperlipidemia, liquid–liquid extraction

## Abstract

Retinoic acids are vitamin A metabolites that have numerous essential functions in humans, and are also used as drugs to treat acne and acute promyelocytic leukemia. All-trans retinoic acid (*at*RA) is the major occurring metabolite of retinoic acid in humans. This study provides a sensitive and specific liquid chromatography–tandem mass spectrometry approach in order to quantify *at*RA in human plasma samples. The isolation of *at*RA by hyperacidified liquid–liquid extraction using hexane and ethyl acetate resulted in a recovery of 89.7 ± 9.2%. The lower limit of detection was 20 pg·mL^−1^, and 7 point calibration displayed good linearity (R^2^ = 0.994) in the range of 50–3200 pg mL^−1^. Selectivity was guaranteed by the use of two individual mass transitions (qualifier and quantifier), and precision and accuracy were determined intraday and interday with a coefficient variation of 9.3% (intraday) and 14.0% (interday). Moreover, the method could be used to isolate *at*RA from hyperlipidemic samples. Applying this method to plasma samples from patients with poorly controlled Type 2 diabetes significantly decreased *at*RA plasma levels as compared to those of the healthy controls. In addition, *at*RA concentrations were highly associated with increased low-density lipoprotein (LDL) and decreased high-density lipoprotein (HDL) cholesterol levels.

## 1. Introduction

Vitamin A is absorbed in the small intestine as retinol, and its metabolites have various physiological functions during development and for the maintenance of cellular homeostasis. Retinol can be oxidized to retinaldehyde, which is essential for vision, and further to retinoic acids, which play key roles in the regulation of the immune system, cell growth, and cell differentiation. Although several different retinoic acids are synthesized in cells, all-trans retinoic acid (*at*RA) is the major active metabolite of retinol, which mediates the majority of physiological functions [1]. *At*RA is a lipophilic substance that acts through binding to DNA-bound nuclear receptors. This allows for the regulation of over 500 genes depending on cell type [2]. Physiological *at*RA levels are maintained by regulated synthesis through aldehyde dehydrogenases and degradation by enzymes of the cytochrome P450 family [3].

In clinical practice, *at*RA is used for the treatment of acne (topical application) and acute promyelocytic leukemia (systemically). In blood, *at*RA undergoes a rapid turnover, and only degradation products, such as 4-oxo-*at*RA, can be detected in urine [4,5]. It is likely that the metabolism of retinoids is affected or may display a causal role in the etiology of various diseases [1,3]. For example, the production rate of *at*RA is regulated by food intake and insulin [6], while *at*RA inhibits progression of diabetic nephropathy in a diabetes mouse model by preventing nuclear translocation of NF-κB in glomeruli and proximal tubules [7]. This indicates the potential role of *at*RA in the pathogenesis of diabetes mellitus and diabetic complications, which warrants further mechanistic investigations and association studies to determine whether the circulatory levels of *at*RA levels would be useful as biomarkers predicting diabetic complications.

Several studies over the last 30 years addressed *at*RA plasma levels in humans using different technical approaches [8,9,10,11,12]. However, such approaches were used for therapeutic drug monitoring in acute promyelocytic leukemia or for determining *at*RA plasma concentrations in healthy volunteers with less specific and sensitive techniques than what is currently possible [8,10,13,14,15,16].

This study aimed at establishing a highly sensitive state-of-the art method in order to quantify plasma concentrations of *at*RA and to apply it to the analysis of patients with Type 2 diabetes. Here, we report a robust and simple extraction assay for *at*RA that can be applied to hyperlipidemic plasma samples, a common preanalytical challenge in laboratory medicine, particularly with respect to obese or diabetic patients [17,18]. The presented method allows for sufficient recovery, and sensitive and specific quantification. In addition, given *at*RA instability, this study aimed to assess the conditions under which *at*RA levels remain constant either in the blood or after extraction.

## 2. Results

### 2.1. Fragmentation and Mass Transitions

The acquisition of *at*RA was achieved in the positive ionization mode of mass spectrometry. In the negative ionization mode of mass spectrometry, the abundance of precursor and product ion was ~10-fold lower than that of positive ionization (Appendix A). The precursor and product ion scan for *at*RA revealed fragments of 283.1 *m*/*z* (qualifier) due to water loss, and 122.9 *m*/*z* (quantifier) caused by the loss of the complete side chain of the carbonyl groups (Figure 1A). This was confirmed by the fragmentation patterns of *at*RA-d5. However, the smaller fragment of 126.8 *m*/*z* (quantifier) displayed a loss in deuterated hydrogen (Figure 1B).

### 2.2. Chromatography

The efficient separation of *at*RA was achieved using a reverse-phase C18 column. The positive ionization of the precursor ions was enhanced by the addition of formic acid. The use of a simple biphasic gradient composed of water and methanol resulted in a chromatogram with a stable peak shape and reproducibility, and a low noise signal (Figure 2C,D). However, to avoid nonspecific peaks (Figure 2C; RT, ~9 min) due to a mild carry-over effect, five consecutive blanks composed of water and methanol (1:1) were run every 50 samples.

### 2.3. Analytical Specificity

The purity verification of *at*RA and *at*RA-d5 was provided by the supplier company. The two product ions with the highest intensity were selected as quantifier and qualifier (Table 1). Using artificial human plasma (Biseko^®,^ Dreieich, Germany) without endogenously produced *at*RA, no coeluting compounds were found, which interfered with the detection of either quantifier or qualifier. However, the possibility that any medications could interfere with our assay could not be excluded. Following the optimization of multiple reaction monitoring (MRM), several injections of spiked artificial plasma showed that the fragmentation patterns for qualifier and quantifier of *at*RA and *at*RA-d5 were reproducible and stable over a long period of time with acceptable accuracy and precision (*n* ≥ 100 injections, Table 2).

### 2.4. Linearity and Determination Limits

For the 7 point calibration, the ratio between *at*RA and *at*RA-d5 was used for quantification. The obtained calibration coefficients (*R^2^*) for 7 point calibration (50–3200 pg·mL^−1^) were > 0.99 (Figure 1A). The achieved lower limits of detection and quantification were 20 and 50 pg·mL^−1^, respectively (Figure 1A,B).

### 2.5. Recovery, Precision, and Accuracy

The extraction recovery of *at*RA-d5 in plasma was 89.7 ± 9.2% (Table 2). Methanol precipitation and acidification with hydrochloric acid resulted in the best recovery as compared to other precipitation or acidification reagents such as trichloroacetic acid, ethanol, acetonitrile, or formic acid (data not shown). The precision of the replicate analyses was evaluated for 50, 600, and 2500 pg·mL^−1^. The coefficient of variation (CV) was intraday at 9.3% and interday at 14.0% (Table 2). Accuracy achieved a mean value of 96.5% on intraday basis and a mean of 101.2% on interday basis (Table 2). The long-term accuracy and precision of artificial plasma (*n* ≥ 100 samples spiked with varying *at*RA concentration) were 94.0 ± 14.2% and 17.8% CV, respectively (Table 2).

### 2.6. Stability

Following storage in darkness under various conditions, samples were quantified and compared to the quantification directly after extraction. The storage of plasma for 1 week at 4 °C was associated with a significant reduction in *at*RA (~51%; Table 3). Postprocessing stability in an assay buffer revealed that *at*RA was degraded when stored for 1 week at 4 °C. Several freeze–thaw cycles were not associated with a significant reduction in *at*RA (~14% reduction; Table 3). Furthermore, exposure to 254 nm UV light for 3 h was associated with ~40% reduction in *at*RA (Appendix A).

### 2.7. Clinical Application

This method was developed to quantify *at*RA in human plasma samples. A total of 14 plasma samples had triglyceride levels above 300 mg·dL^−1^. The best recovery in those samples was achieved following strong acidification with HCl (10 M), liquid–liquid extraction with a combination of ethyl acetate/hexane (1:1), and the incubation of the sample at 4 °C for 20 min. No acidification and the usage of different extraction solvents were associated with poor recovery in those hyperlipidemic samples (data not shown). For a constant extraction procedure, this method was applied to all plasma samples.

Using the developed LC–MS/MS method, plasma levels of *at*RA were found to be in the range of 0.57 to 2.76 ng·mL^−1^ in all patients. A cohort of patients with poorly controlled Type 2 diabetes displayed hyperlipidemia, increased C-reactive protein, and blood glucose concentration, as well as excessively increased HbA1c (Table 4). In patients with poorly controlled Type 2 diabetes, *at*RA levels were significantly lower, with 1.38 ± 0.42 ng·mL^−1^, as compared to healthy individuals with 1.77 ± 0.45 ng·mL^−1^ (*p* < 0.01; Figure 3A). In addition, plasma *at*RA levels were strongly associated with the lipid profiles of the whole cohort (*n* = 44), especially for high-density lipoprotein (HDL; r = 0.598; *p* < 0.001) and low-density lipoprotein (LDL) cholesterol (r = −0.469; *p* < 0.01), and also for body-mass index (BMI) and triglyceride concentration (Figure 3B–F). In a cohort of patients with well-adjusted Type 2 diabetes, *at*RA levels decreased (1.47 ± 0.37 ng·mL^−1^), but this was statistically not significant (*p* = 0.053 vs. controls; Appendix A).

## 3. Discussion

This study offers a sensitive and specific liquid chromatography, tandem mass spectrometry (LC–MS/MS) approach for the quantification of *at*RA in hyperlipidemic human plasma samples. Within this context, this method for the first time describes the usage of 122.9 *m*/*z* for the quantification of *at*RA in biological matrices, which may be a result of a more efficient fragmentation in the collision cell as compared to that of other studies [6,8,19]. As previously described, the separation and ionization of *at*RA were achieved using a reverse-phase C18 column and formic acid [8,9,14,20]. With 20 and 50 pg·mL^−1^ as the lower limits of detection and quantification, this method is more sensitive than other published methods in the field that use LC–MS and high-performance liquid chromatography (HPLC) approaches [9,10,11,15,16]. Ex vivo, *at*RA is chemically unstable, and readily oxidized and isomerized when exposed to heat, light, or oxygen [15,16], which reflects the necessity for stability analysis. Pre- and postprocessing analyses revealed that *at*RA had sufficient short-term stability, but degraded when stored for 1 week at 4 °C, which is in line with previous studies on *at*RA stability [21]. Exposure to 254 nm UV light for 3 h or longer was associated with a significant reduction in *at*RA. Therefore, the usage of amber reaction tubes and the avoidance of direct sunlight during analysis are strongly recommended.

As a lipophilic compound, *at*RA presents challenges during extraction, particularly in patients with an altered lipid profile displayed by increased triglycerides, LDL, and total cholesterol. With plasma levels of *at*RA between 0.57 and 2.76 ng·mL^−1^ in all patients, our results were in line with those of other studies in healthy individuals, where 0.3–5 ng·mL^−1^ of *at*RA was found to be present in the plasma and serum [8,9,10,15,16]. In patients with poorly controlled Type 2 diabetes, lower *at*RA levels than those of healthy individuals were found. However, a cohort of patients with well-adjusted Type 2 diabetes displayed only a minor decrease in *at*RA. Future studies should address whether *at*RA could be causally involved in the progression of diabetes mellitus.

In all analyzed individuals, the plasma *at*RA levels were strongly associated with lipid profiles and body-mass index (BMI). This confirms a recent study in which *at*RA was claimed to be a predictive marker for the development of metabolic syndrome, which is strongly associated with the development of Type 2 diabetes and cardiovascular disease [22]. Experimental diabetic model systems showed that *at*RA can block adipocyte differentiation [23]. Furthermore, *at*RA affects insulin secretion, which is not surprising given the ubiquitous expression of the retinoic X receptor [24]. Whether the reduction in *at*RA plasma concentration in dyslipidemic patients is the cause or effect of lipid dysregulation remains unclear and should be addressed in further studies.

## 4. Materials and Methods

### 4.1. Chemicals and Reagents

Acetonitrile, hexane, methanol, hydrochloric acid (37%), and water were of high purity grade and purchased from Sigma–Aldrich (Steinheim, Germany). Formic acid was purchased from Biosolve (Valkenswaard, Netherlands). All-trans retinoic acid (*at*RA) and all-trans retinoic acid-d5 (*at*RA-d5) were purchased as the LC–MS standard (purity > 98%) from Biomol (Hamburg, Germany). Amber reaction vials (1.5 mL) were purchased from Chromsystems (Gräfelfing, Germany). Artificial plasma with a standardized protein composition of human plasma (Biseko^®,^ Dreieich, Germany) was purchased from Biotest (Dreieich, Germany) and used as a matrix for method development and validation. Biseko^®^ is a virus-inactivated human-plasma substitute that contains the entire spectrum of serum proteins in a standardized active form [25]. It is prepared from plasma pools of at least 1000 individual donations. One liter of Biseko^©^ solution contains 50 g of total protein, including albumin (31 g), IgG (7.1 g), IgA (1.55 g), IgM (0.48 g), sodium ions (3.56 g), potassium ions (0.16 g), calcium ions (0.08 g), magnesium ions (0.02 g), and chloride ions (3.65 g).

### 4.2. Preparation of Calibration Standards

Stock solutions of *at*RA and *at*RA-d5 were prepared in dimethyl sulfoxide at concentrations of 500 (*at*RA) or 100 µg·mL^−1^ (*at*RA-d5) and stored at −80 °C. Working solutions of *at*RA were prepared in methanol in the range of 500–100 µg·mL^−1^ and stored at −20 °C. Calibration standards of *at*RA were 50, 100, 200, 400, 800, 1600, and 3200 pg·mL^−1^ with the addition of 500 pg·mL^−1^ of *at*RA-d5 for each calibrator. All stock and working solutions were kept in amber reaction vials to avoid any degradation by ultraviolet light.

### 4.3. Sample Collection and Extraction

Ethylenediaminetetraacetic acid (EDTA) plasma samples were obtained from 20 healthy controls and 39 patients with Type 2 diabetes. All participants were in a fasting state, and patients with Type 2 diabetes were enrolled if diagnosis had been established according to the guidelines of the German Diabetes Association [26]. The study was approved by the ethics committee of Heidelberg University Hospital (Project Identification Code S-383/2016). All patient material and data were acquired with formal written informed consent and in agreement with the guidelines of the ethics committee as previously described [19]. The plasma supernatant was directly aliquoted following centrifugation (5 min at 5000× *g*; 4 °C) of the whole blood and frozen at −80 °C. Samples were thawed once, and liquid–liquid extraction was carried out under reduced light condition in amber tubes. The internal standard (100 pg of *at*RA-d5) was added to 200 µL of plasma, which was acidified with 5 µL of hydrochloric acid (10 M). Afterwards, 400 µL of methanol was added for protein precipitation. After centrifugation (10 min at 10,000× *g*; 4 °C), the supernatant was transferred into a new amber tube and 300 µL of hexane and 300 µL of ethyl acetate was added, which was then mixed for 10 s. The mixture was left for 20 min at 4 °C in darkness, and the aqueous and organic phase was separated by centrifugation (10 min at 10,000× *g*; 4 °C). The upper organic phase was evaporated using a vacuum concentrator (Eppendorf Concentrator Plus) at room temperature. The residue was resuspended in 60 µL of methanol/water (2:1) and transferred into an HPLC vial (Waters, Eschborn, Germany).

### 4.4. Chromatography

All analyses were performed on a Waters^®^ Acquity UPLC class I system (Waters, Eschborn, Germany) equipped with a binary solvent-delivery system with an online degasser and a column manager containing a column oven connected to a UPLC autosampler. *At*RA was separated by reverse-phase LC on a Waters^®^ Acquity BEH C18 column (1.7 µM, 2.1 × 50 mm) at a flow rate of 0.2 mL·min^−1^ and a column temperature of 20 °C. During analyses, all samples were stored in the autosampler at a temperature of 4 °C, and the injection volume for each sample varied between 10 and 30 µL. Solvent A consisted of 0.1% formic acid in water, and Solvent B was 0.1% formic acid in methanol. For each run, gradient elution was performed. Solvent A was decreased from 80% to 0% (0–4 min), remained isocratic at 0% (4–6 min), and increased back to 80% (6–15 min). The column eluent was directed into the MS, and analyses were performed using MassLynx XS software.

### 4.5. Mass Spectrometry

*At*RA detection was performed on a XEVO TQ-S tandem quadrupole mass spectrometer (Waters^®^) equipped with an electrospray ionization source (ESI) operated in positive ion mode. Analyte detection was performed using multiple reaction monitoring (MRM). Source parameters were set as follows: capillary voltage, 3.5 kV; desolvation temperature, 350 °C; desolvation gas flow, 850 L·h^−1^; source temperature, 200 °C; cone gas flow, 250 L·h^−1^; collision gas flow, 0.15 mL·min^−1^; nebulizer gas flow, 5 bar. Cone and collision voltage were individually optimized for *at*RA and *at*RA-d5 and are summarized in Table 1. Acquisition and quantification were completed with MassLynx 4.1 and TargetLynx 2.7.

### 4.6. Validation Procedure

The method was validated for selectivity, matrix effects, linearity, lower limits of detection (LLOD) and quantification (LLOQ), recovery, stability, precision, and accuracy (intra/interday). Seven-point calibration was performed using linear regression. For all concentration calculations, the area ratio of *at*RA/*at*RA-d5 was plotted against nominal calibrator concentration. Blank samples (without analytes) were measured for each calibration curve to ensure reliability. For the determination of recovery, LLOD, LLOQ, precision, and accuracy, 200 µL of spiked artificial plasma was used. Recovery, precision, and accuracy were validated in an intraday assay using three different concentrations (50, 600, 2500 pg·mL^−1^), which were measured in quadruplicates. For interday variability, the same concentrations (50, 600, 2500 pg·mL^−1^) of spiked artificial plasma were quantified on 3 consecutive days in quadruplicates. LLOD and LLOQ were determined with a signal-to-noise ratio of 6.4 (LLOQ) and 3.2 (LLOD). Stability was validated in human artificial plasma (preprocessed) and in assay buffer (postprocessed) at various temperature levels and for different durations by spiking 200 µL of plasma sample with 1000 pg of atRA. *At*RA stability was also assessed for exposure to UV light (254 nm) using transparent tubes. Matrix effects were defined as a suppression of or increase in signal intensity for the chosen MRMs (matrix effects while ionization) or as an increase or decrease in recovery of atRA-d5 (matrix effect while extraction).

### 4.7. Clinical Chemistry

Blood was drawn under fasting conditions as described above, and samples were immediately processed in the accredited Central Laboratory of Heidelberg University Hospital. Plasma levels of total cholesterol, HDL cholesterol, triglycerides, blood glucose, and C-reactive protein were analyzed with clinical chemistry automation (AdviaXPT^®^ 2400 chemistry analyzer, Siemens Healthineers, Erlangen, Germany) according to the appropriate standard operating protocol. HbA1c quantification was performed on whole-blood EDTA samples, and analyses were executed using a VARIANT II HbA1c analyzer (BioRad, Hercules, CA, USA) that utilizes cation-exchange high-performance liquid chromatography. LDL cholesterol was determined using the Friedewald formula as previously described [27].

### 4.8. Statistical Analysis

Statistical-data analysis was performed using GraphPad Prism 7 (GraphPad Software Inc., San Diego, CA, USA). All data are expressed as mean values ± standard error (SE), and were analyzed for significance using an unpaired t test with Welch’s correction. Linear regression and Pearson’s correlation coefficient were used to study the association of *at*RA and BMI, total cholesterol, LDL cholesterol, HDL cholesterol, and triglycerides.

## 5. Conclusions

In this study, *at*RA was quantified in human-plasma samples using a robust and sensitive LC–MS/MS method that achieved convincing analytical parameters of accuracy, precision, recovery, and stability. Within this context, *at*RA was isolated from hyperlipidemic human-plasma samples, and results showed a decrease in *at*RA levels in poorly controlled patients with Type 2 diabetes, and a strong correlation with LDL and HDL cholesterol. In summary, this study adds valuable content for further analytical determinations of *at*RA using LC–MS/MS, and indicates altered metabolism of retinoids in patients with diabetes mellitus.

## Figures and Tables

**Figure 1 metabolites-11-00060-f001:**
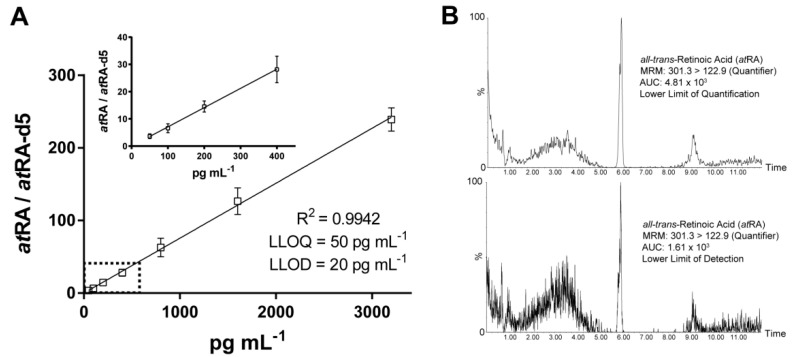
Parameters of linearity and limitations. (**A**) Calibration range (50-3200 pg·mL^−1^); lower limit of quantification (LLOQ) (signal-to-noise ratio, 6.4/1); lower limit of detection (LLOD; signal-to-noise ratio, 3.2/1) and coefficient of determination (R2). (**B**) Total ion chromatogram of all-trans retinoic acid (*at*RA) for LLOQ (top) and LLOD (bottom).

**Figure 2 metabolites-11-00060-f002:**
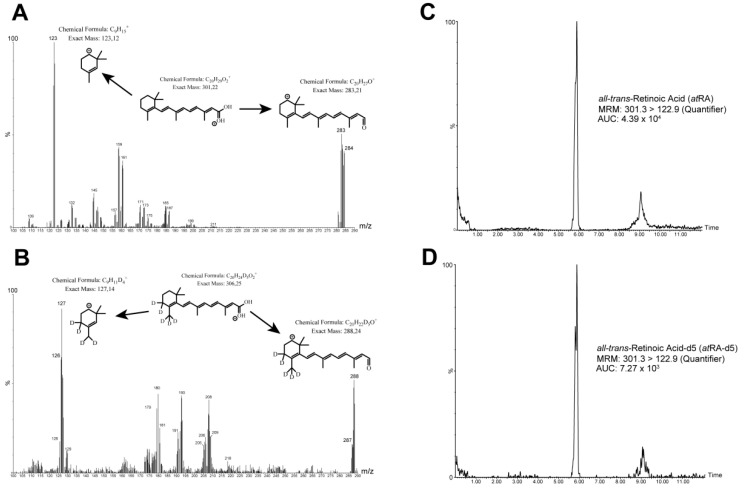
Multiple reaction monitoring (MRM) chromatograms and total ion chromatograms of *at*RA/*at*RA-d5. (**A**) Extracted MRM spectrum (quantifier/qualifier) of *at*RA. (**B**) Extracted MRM spectrum (quantifier/qualifier) of atRA-d5. (**C**) Total ion chromatogram of *at*RA in a control subject (1.02 ng·mL^−1^). (**D**) Total ion chromatogram of *at*RA-d5 in the same control subject as in (**C**).

**Figure 3 metabolites-11-00060-f003:**
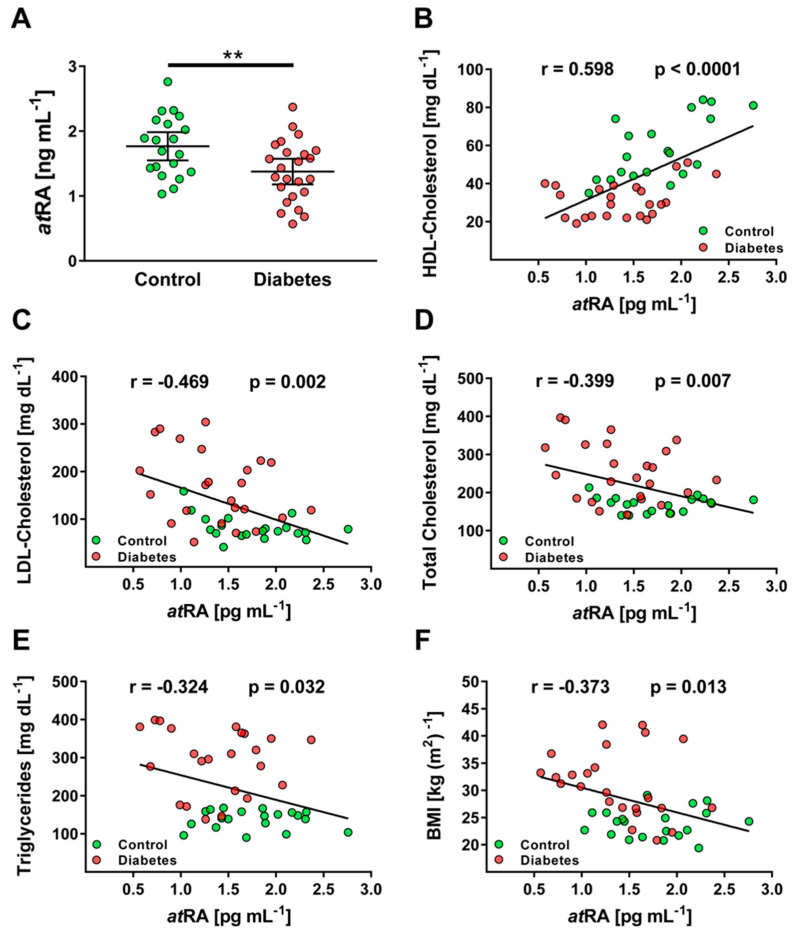
Clinical application and associations with lipid profile in human plasma samples. (**A**) Quantification of *at*RA in control patients (*n* = 20) and patients with Type 2 diabetes (*n* = 24). Correlation analysis of (**B**) *at*RA and HDL cholesterol, (**C**) *at*RA and LDL cholesterol, (**D**) *at*RA and total cholesterol, (**E**) *at*RA and triglycerides, and (**F**) *at*RA and BMI. (**B**–**F**) Correlation analysis was performed in the whole cohort. Level of significance and Pearson’s correlation coefficient (r) are shown. Data are mean ± SEM. ** *p* < 0.01.

**Table 1 metabolites-11-00060-t001:** Retention time (Rt), mass transition (MRM), cone voltage (COV), and collision energy (CE) for *at*RA and *at*RA-d5.

Analytes	R_t_ (min)	MRM Quantifier (*m*/*z*)	MRM Qualifier (*m*/*z*)	COV (V)	CE (V)
*at*RA	5.90	301.3 > 122.9	301.1 > 283.1	23	16
*at*RA-d5	5.89	306.3 > 126.8	306.3 > 288.1	26	16

**Table 2 metabolites-11-00060-t002:** Parameters of variability and extraction recovery for *at*RA. Intra- and interday accuracy/precision and extraction efficiency based on recovery of *at*RA-d5. Accuracy is the mean of quantified concentration given as percentage of three spiked concentrations (nominal) in human artificial plasma samples. Precision is the coefficient of variation (CV) of mean concentration determined for three different concentration levels (intra/interday, *n* = 4). The precision of retention time is described as the CV of the mean retention time of 120 injections (21 calibrators, 99 samples) over a period of 65 days. Long-term accuracy and precision were determined over a period of 30 days with 105 injections of varying concentration within the calibration range.

Parameter	Nominal Concentration (pg mL^−1^)	Recovery (%)	Measured Concentration (pg mL^−1^)	Accuracy (%)	Precision (% CV)	Precision of Retention Time (% CV)
Intraday	50 600 2500	87.4 ± 8.2 90.0 ± 6.4 91.7 ± 12.9	46.8 ± 4.6 618.6 ± 41.7 2320.4 ± 263.2	93.6 103.1 92.8	9.8 6.7 11.3	5.5
Interday	50 600 2500		57.9 ± 8.3 519.4 ± 81.2 2549.6 ± 307.3	115.8 86.6 102.0	14.3 15.6 12.1	
Long-term	Various			94.0 ± 14.2	17.8	

**Table 3 metabolites-11-00060-t003:** Pre- and postprocessed sample stability of *at*RA under varying conditions. Stability is a change in percentage calculated by measured concentrations divided by nominal concentrations (samples were spiked with 1000 pg·mL^−1^ of atRA; *n* = 4 for each test).

Conditions	Recovery	
	Preprocessed	Postprocessed
1 h at 20 °C	93 ± 12%	
6 h at 20 °C	84 ± 19%	
1 h at 4 °C	89 ± 17%	
6 h at 4 °C	81 ± 21%	77 ± 18%
1 week at 4 °C	51 ± 31%	44 ± 14%
1 week at −20 °C		91 ± 9%
1 month at −20 °C	101 ± 15%	93 ± 16%
F/T stability (6 cycles)		82 ± 22%

**Table 4 metabolites-11-00060-t004:** Mean baseline characteristics of control and diabetic patient cohorts. All parameters were determined prior to collection. Data are mean ± SD; *** *p* < 0.001, vs. controls. All other characteristics were not significant (*p* > 0.05).

Parameter-	Controls (*n* = 20)	Type 2 Diabetes (*n* = 24)
Sex (% male)	63.8	65.3
Age (years)	42.1 ± 12.3	49.6 ± 15.5
Body-mass index (BMI; kg·m^−2^)	24 ± 2.5	31.3 ± 6.1 ***
Blood glucose (mg·dL^−1^)	97.1 ± 12.3	186.3 ± 84.3 ***
HbA1c (%)	5.7 ± 0.8	11.2 ± 1.6 ***
C-reactive protein (CRP; mg·L^−1^)	1.2 ± 1.1	12.9 ± 8.6 ***
Total cholesterol (mg dL^−1^)	168.4 ± 19.9	256.1 ± 74.5 ***
Triglycerides (mg dL^−1^)	137.7 ± 24.2	285.3 ± 86.4 ***
Low-density lipoprotein (LDL; mg·dL^−1^)	82.7 ± 25.7	167.5 ± 73.4 ***
High-density lipoprotein (HDL; mg·dL^−1^)	58.2 ± 15.9	31.5 ± 9.1 ***

## Data Availability

The data presented in this study are available on request from the corresponding author. The data are not publicly available due to privacy and/or ethical concerns.

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
