# Peer review of "Quantification of All-Trans Retinoic Acid by Liquid Chromatography–Tandem Mass Spectrometry and Association with Lipid Profile in Patients with Type 2 Diabetes"

_metabolites, 2021, doi:10.3390/metabo11010060_

Round 1

Reviewer 1 Report

Review

In the manuscript entitled “Quantification of all-trans retinoic acid by liquid chromatography-tandem mass spectrometry and association with lipid 3 profile in patients with type 2 diabetes,” the authors evaluated the performances of acidic liquid-liquid extraction of all-trans retinoic acid and a newly developed LC-MS/MS method. The latter utilized the tandem mass spectrometry in the positive ion mode to obtain quantifier and qualifier fragment ions, which were confirmed via deuterated analogs. The established approach yielded good metabolite extraction recovery, good analytical accuracy, and acceptable analytical precision across three different concentrations of all-trans retinoic acid spiked plasma. The analytical method achieved pg mL-1 levels of detection with good linearity. This approach was used to evaluate all-trans retinoic acid levels in patients with type 2 diabetes.

Comments:

  1. In line 70, please change “levels are remain constant” to “levels remain constant”.
  2. In line 73, please consider revising to “Acquisition of atRA was achieved in positive ionization mode of mass spectrometry”.
  3. For all tables, please use capital letters for the headings of each column =(e.g., Intraday, Analytes, Measured Concentration [pgmL-1], Recovery [%], Pre-processed, etc.).
  4. In lines 94 to 97, the authors state “Following the optimization of multiple reaction monitoring (MRMs), several injections of spiked artificial plasma showed that fragmentation patterns for qualifier and quantifier of atRA and atRA-d5 were reproducible and stable over a long period of time (n > 100 injections)”. The authors should provide accuracy and precision measurements for the >100 injections. This would be very informative. Perhaps this data can be added to Table 2 or simply stated in the text.
  5. In line 97, please change to (n = > 100 injections).
  6. In Table 2, the authors should provide chromatographic retention time precision measurements (intraday and interday) as well.
  7. The authors should consider rewording the title of Table 3.
  8. In line 164, please change “(C) total ion chromatogram” to “(C) Total ion chromatogram”.
  9. In lines 146 and 149, the authors are referring to Figure 3 A and Figure 3 B – F, respectively. The authors should clarify this.
  10. Did the authors try their LC-MS/MS method in the negative ion mode of mass spectrometry? If so, how did it compare the positive ion mode?
  11. In line 280, the authors state that a six-point calibration curve was performed but a seven-point calibration curve is described in the rest of the manuscript. Can the authors clarify this?

Author Response

The authors would like to thank the reviewer for the very good rating of this manuscript and for providing us with constructive criticism as well as suggestions to improve the quality of this manuscript.

Reviewer #1 (Specific Comments):

  1. In line 70, please change “levels are remain constant” to “levels remain constant”.

Response:

The text has been corrected accordingly.

  1. In line 73, please consider revising to “Acquisition of atRA was achieved in positive ionization mode of mass spectrometry”..

Response:

The text has been corrected accordingly.

  1. For all tables, please use capital letters for the headings of each column =(e.g., Intraday, Analytes, Measured Concentration [pgmL-1], Recovery [%], Pre-processed, etc.).

Response:

The headings of each column in all tables have been changed to capital letters.

  1. In lines 94 to 97, the authors state “Following the optimization of multiple reaction monitoring (MRMs), several injections of spiked artificial plasma showed that fragmentation patterns for qualifier and quantifier of atRA and atRA-d5 were reproducible and stable over a long period of time (n > 100 injections)”. The authors should provide accuracy and precision measurements for the >100 injections. This would be very informative. Perhaps this data can be added to Table 2 or simply stated in the text.

Response:

The authors agree that the data would be informative and have added an appropriate section to Table 2 and a short description to the results section "2.5 Recovery, Precision and Accuracy".

  1. In line 97, please change to (n = > 100 injections).

Response:

The text has been corrected accordingly.

  1. In Table 2, the authors should provide chromatographic retention time precision measurements (intraday and interday) as well.

Response:

The authors have added the precision data of the retention time to Table 2. It summarizes the coefficient of variation for 120 injections which includes 21 calibrator and 99 sample injections over a period of 65 days.  

  1. The authors should consider rewording the title of Table 3.

Response:

The title of Table 3 has been rephrased.

  1. In line 164, please change “(C) total ion chromatogram” to “(C) Total ion chromatogram”.

Response:

The text has been corrected accordingly.

  1. In lines 146 and 149, the authors are referring to Figure 3 A and Figure 3 B – F, respectively. The authors should clarify this.

Response:

To clarify, the authors had incorrectly referred to Figure 4, which does not exist. The text has been corrected accordingly.

  1. Did the authors try their LC-MS/MS method in the negative ion mode of mass spectrometry? If so, how did it compare the positive ion mode?

Response:

The authors used the approach of negative ion mode mass spectrometry when tried to establish the MRM for atRA. The intensity was approximately 10-fold lower as compared to positive ionization. However, the authors agree that this is an important result for other researchers in the field and added a supplementary figure illustrating this phenomenon (Supplementary Figure 1). Additionally, a short paragraph referring to this observation has been added to the section "2.1 Fragmentation and Mass Transition".

  1. In line 280, the authors state that a six-point calibration curve was performed but a seven-point calibration curve is described in the rest of the manuscript. Can the authors clarify this?

Response:

To clarify, a seven-point calibration curve was performed. The text has been corrected accordingly.

Reviewer 2 Report

There is publications of 1) atRA quantification using LCMS, 2) change of atRA in diabetes.

Therefore, without any specification of this study from all previous works, it looks like different to claim any novelty of this study. 

Here are my comments:

The authors established quantitative LCMS assay for all-trans retinoic acid and determined the levels of all-trans retinoic acids in control and diabetes patients.
Although they established robust method for measurement of blood atRA levels and lowered the detection range, the determined range of atRA in patients is covered by previously established method by other groups. This indicate that new established method by authors is not critical for determination of blood atRA levels of patients in the present study. In addition, the number of subjects in the present study were limited (just 20-24 subjects in group) without any matching of BMI, CRP, lipid parameters. This limitation making complicated what factor associated with the differences in atRA levels and could not lead to any feasible conclusion.
Moreover, even this paper established fine method, the establishment of bioanalytical method itself is not scope of metabolites.
Therefore, I suggest submission of this paper to other journal, which mainly accept bioanalytical method, instead of Metabolites.

Author Response

There is publications of 1) atRA quantification using LCMS, 2) change of atRA in diabetes.

Therefore, without any specification of this study from all previous works, it looks like different to claim any novelty of this study.

The authors established quantitative LCMS assay for all-trans retinoic acid and determined the levels of all-trans retinoic acids in control and diabetes patients.

Response:

The authors would like to thank the reviewer for providing us with constructive criticism and suggestions to improve the quality of this manuscript. The authors agree that previous studies have addressed the quantification of atRA using LCMS and (partly) also the levels of atRA in diabetes. However, taking into consideration the variety and complexity of different LC-MS devices and their adjustable parameters for each metabolite, our study adds valuable content regarding the method development. Additionally, within the context of extraction, this study describes the isolation of atRA from hyperlipidemic samples for the first time.        

Reviewer #2 (Specific Comments):

  1. Although they established robust method for measurement of blood atRA levels and lowered the detection range, the determined range of atRA in patients is covered by previously established method by other groups. This indicate that new established method by authors is not critical for determination of blood atRA levels of patients in the present study.

Response:

The authors agree that this study confirms other bioanalytical approaches which quantified atRA in human blood samples. The relevant studies were cited and discussed. However, when it comes to the translation of valuable metabolites/biomarkers from research to routine diagnostics, the evaluation of different approaches for quantification, which have been determined by comparable techniques, is essential in order to establish reference ranges and prove principles vice versa.

As stated by the reviewer correctly, our study provides this comparable technique in a robust and sensitive fashion with a broad analytical range from 50 - 3200 pg mL-1. Using this detection method we also present a novel approach to extract atRA in an inexpensive manner from hyperlipidemic plasma samples. Hyperlipidemic samples are a common preanalytical challenge in laboratory medicine, particularly with respect to diabetic patients. Clinical researchers potentially benefit from our method not only for the quantification of atRA, but also other lipophilic compounds.

  1. In addition, the number of subjects in the present study were limited (just 20-24 subjects in group) without any matching of BMI, CRP, lipid parameters. This limitation making complicated what factor associated with the differences in atRA levels and could not lead to any feasible conclusion.

Response:

The authors are aware of this and agree that the differences in atRA could be a potential selection bias in our study cohort. Studying the plasma levels of atRA in a cohort with poorly controlled diabetes (HbA1c above 10%) is a novelty itself. Unfortunately, it is nearly impossible to find a perfectly matched control cohort regarding BMI, CRP and lipid profile. This is a result of the fact that individuals from this diabetes cohort are obese which undoubtedly influences the BMI, CRP and lipid profile. Nevertheless, the number of patients was sufficient to achieve significant differences regarding atRA plasma levels (p<0.01) as compared to age- and gender-matched controls.

As suggested by the reviewer, the authors have added an additional measurement of 15 patients with diagnosed, but well-adjusted type 2 diabetes, which are matched for BMI, CRP and lipid profile in order to provide a better overview regarding the atRA levels in patients with type 2 diabetes in general. Interestingly, this cohort of well-adjusted type 2 diabetic patients showed also a decrease in atRA plasma concentrations (supplementary Table S2; p = 0.053). This finding might inspire further research to address whether this could be cause or consequence of metabolic dysfunction.

In the revised version of this manuscript we emphasized this finding in the results and discussion, but to unravel responsible factors and possible mechanisms, which lead to altered atRA in those two cohorts, is beyond the scope of this manuscript.

  1. Moreover, even this paper established fine method, the establishment of bioanalytical method itself is not scope of metabolites.

Therefore, I suggest submission of this paper to other journal, which mainly accept bioanalytical method, instead of Metabolites.

Response:

The authors understand to some degree the reviewer’s reservations regarding the novelty of this study (see above), but do not understand why this study is not within the scope of the journal "Metabolites". Particularly the special issue "Mass Spectrometry-Based Lipidomics" is encouraging authors to provide manuscripts which explore for example sample preparation, chromatographic separation, MS and MS/MS, and clinical translation. Furthermore, the journal "Metabolites" has just recently published articles of high quality describing an LC-MS methods and their clinical translation (e.g.: 10.3390/metabo10120495; 10.3390/metabo10120494; 10.3390/metabo11010011; 10.3390/metabo10110464).      

Reviewer 3 Report

This is an excellent report on the MS-based measurement of an important metabolite, all-trans retinoic acid in human plasma samples with clinical relevance. I found the article easy to navigate and the data are very well prepared and presented.

Minor points:

  1. The formatting of the print layout can improved (tables across two pages).
  2. The language needs minor correction (e.g. 'The method was used' instead of 'could be used')

Author Response

This is an excellent report on the MS-based measurement of an important metabolite, all-trans retinoic acid in human plasma samples with clinical relevance. I found the article easy to navigate and the data are very well prepared and presented.

Response:

The authors would like to thank the reviewer for thoroughly reading our manuscript and providing us with such an excellent rating.

Reviewer #3 (Specific Comments):

  1. The formatting of the print layout can improved (tables across two pages).

Response:

The authors have rearranged the layout of the tables to achieve a better overview regarding tables and figures.

  1. The language needs minor correction (e.g. 'The method was used' instead of 'could be used')

Response:

The authors have performed a language check including corrections of grammar and spelling. All changes are highlighted in the revised manuscript.

Round 2

Reviewer 2 Report

All comments were well addressed.

In addition, once following author response is clarified with data, this paper would be acceptable for publication.

"Using this detection method we also present a novel approach to extract atRA in an inexpensive manner from hyperlipidemic plasma samples. Hyperlipidemic samples are a common preanalytical challenge in laboratory medicine, particularly with respect to diabetic patients."

Author Response

All comments were well addressed.

In addition, once following author response is clarified with data, this paper would be acceptable for publication.

"Using this detection method we also present a novel approach to extract atRA in an inexpensive manner from hyperlipidemic plasma samples. Hyperlipidemic samples are a common preanalytical challenge in laboratory medicine, particularly with respect to diabetic patients."

Response:

The authors would like to thank the reviewer for the good rating and the possibility for eventual acceptance. The authors added a statement to the last part of the introduction about preanalytical challenges of hyperlipidemic samples in laboratory medicine, which is supported by the cited literature (line 68-69, References 17, 18].